# Maternal smoking behaviour across the first two pregnancies and small for gestational age birth: Analysis of the SLOPE (Studying Lifecourse Obesity PrEdictors) population-based cohort in the South of England

Elizabeth J. Taylor[1,2]*, Pia Doh[1], Nida Ziauddeen[1], Keith M. Godfrey[2,3], Ann Berrington[4], Nisreen A. Alwan[1,2,5]*

1 School of Primary Care, Population Sciences and Medical Education, Faculty of Medicine, University of Southampton, Southampton, United Kingdom, 2 NIHR Southampton Biomedical Research Centre, University of Southampton and University Hospital Southampton NHS Foundation Trust, Southampton, United Kingdom, 3 MRC Lifecourse Epidemiology Unit, University of Southampton, Southampton General Hospital, Southampton, United Kingdom, 4 Department of Social Statistics and Demography, University of Southampton, Southampton, United Kingdom, 5 NIHR Applied Research Collaboration (ARC) Wessex, Southampton, United Kingdom

* E.J.Taylor@soton.ac.uk (EJT); N.A.Alwan@soton.ac.uk (NAA)

**Data Availability Statement:** The study's ethical approval from the Faculty of Medicine Ethics

## Abstract

Maternal smoking is established to cause adverse birth outcomes, but evidence considering maternal smoking change across successive pregnancies is sparse. We examined the association between self-reported maternal smoking during and between the first two pregnancies with the odds of small for gestational age (SGA) birth (<10th percentile) in the second infant.

Records for the first two pregnancies for 16791 women within the SLOPE (Studying Lifecourse Obesity PrEdictors) study were analysed. This is a population-based cohort of prospectively collected anonymised antenatal and birth healthcare data (2003–2018) in Hampshire, UK. Logistic regression was used to relate maternal smoking change to the odds of SGA birth in the second infant.

In the full sample, compared to never smokers, mothers smoking at the start of the first pregnancy had higher odds of SGA birth in the second pregnancy even where they stopped smoking before the first antenatal appointment for the second pregnancy (adjusted odds ratio (aOR) 1.50 [95% confidence interval 1.10, 2.03]). If a mother was not a smoker at the first antenatal appointment for either her first or her second pregnancy, but smoked later in her first pregnancy or between pregnancies, there was no evidence of increased risk of SGA birth in the second pregnancy compared to never smokers. A mother who smoked ten or more cigarettes a day at the start of both of her first two pregnancies had the highest odds of SGA birth (3.54 [2.55, 4.92]). Women who were not smoking at the start of the first pregnancy but who subsequently resumed/began smoking and smoked at the start of their second pregnancy, also had higher odds (2.11 [1.51, 2.95]) than never smokers.

Committee, University of Southampton and the Health Research Authority restricts public sharing of the raw data used in this study. To request access conditional on approval from the appropriate institutional ethics, research governance processes and data owners, please email rgoinfo@soton.ac.uk.

**Funding:** This research is supported by an Academy of Medical Sciences and Wellcome Trust grant to NAA [Grant no: AMS_HOP001\1060] (https://acmedsci.ac.uk/) (https://wellcome.ac.uk) and an NIHR Southampton Biomedical Research Centre PhD studentship to EJT. The funders had no role in study design, data collection and analysis, decision to publish, or preparation of the manuscript.

**Competing interests:** The authors have read the journal's policy and the authors of this manuscript have the following competing interests: EJT is supported by an NIHR Southampton Biomedical Research Centre and University of Southampton Primary Care, Population Sciences and Medical Education PhD studentship. Within 5 years of conducting this study, NAA has received research grants as PI from the Academy of Medical Sciences/Wellcome Trust and the National Institute for Health Research (NIHR) Southampton Biomedical Research Centre and the NIHR Applied Research Collaboration, Wessex. NAA has acted as a member of the National Institute for Health and Care Excellence (NICE) Antenatal Care Guideline Committee. KMG has received reimbursement for speaking at conferences sponsored by companies selling nutritional products and is part of an academic consortium that has received research funding from BenevolentAI Bio Ltd, Abbott Nutrition, Nestec and Danone. KMG is supported by the UK Medical Research Council (MC_UU_12011/4), the National Institute for Health Research (NIHR Senior Investigator (NF-SI-0515-10042), NIHR Southampton 1000DaysPlus Global Nutrition Research Group (17/63/154) and NIHR Southampton Biomedical Research Centre (IS-BRC-1215-20004)), the European Union (Erasmus+ Programme Early Nutrition eAcademy Southeast Asia-573651-EPP-1-2016-1-DE-EPPKA2-CBHE-JP and ImpENSA 598488-EPP-1-2018-1-DE-EPPKA2-CBHE-JP), the British Heart Foundation (RG/15/17/3174), the US National Institute On Aging of the National Institutes of Health (Award No. U24AG047867) and the UK ESRC and BBSRC (Award No. ES/M00919X/1). AB has received funding from the European Commission (Horizon 2020), UKRI(ESRC) and the National Institute for Health Research. The other authors have no potentially competing interests to

Smoking in the first pregnancy was associated with SGA birth in the second pregnancy, even if the mother quit by the confirmation of her second pregnancy.

## Introduction

Maternal smoking has been associated with the inability to conceive as well as the risks of ectopic pregnancy, miscarriage, stillbirth and prematurity [1, 2] and the association between smoking during pregnancy and fetal growth restriction is considered to be causal [1]. A dose response relationship has been shown between the number of cigarettes smoked a day in pregnancy and the risk of placental abruption and negative birth outcomes [1, 3, 4]. The greatest morphological effects in the placenta are found where there is heavy smoking before 10 weeks gestation (> 20 cigarettes a day) [2]. In addition to being born prematurely [5], adverse health consequences for the child include being born small for gestational age (SGA) (<10th percentile) [6] and an increased risk of congenital malformations, primarily oral-facial clefts [7].

A recent systematic review and meta-analysis has estimated that nearly 2% of women globally smoke during pregnancy, with nearly three-quarters of these smoking daily [8]. There is substantial variation between the countries considered in this study with the highest estimated prevalence being in Ireland (38.4% [95% CI [25.4, 52.4]), Uruguay (29.7% [16.6, 44.8]) and Bulgaria (29.4% [26.6, 32.2]) [8]. Figures for the third quarter of 2019/20 show that in England, where this study is based, 10.5% of women report smoking at the time of delivery, although there is substantial regional variation between the lowest and highest rates (from 1.6% in Central London to 23.3% in Blackpool) [9].

Since longitudinal data are sparse, most studies are only able to consider the association between maternal exposures, such as smoking, in one pregnancy with the outcome for that pregnancy, and biological links during the same pregnancy are already established. Few studies have sought to categorise maternal smoking behaviour across successive pregnancies to examine whether the association between SGA and history of smoking extends beyond the period of the same pregnancy or whether exposure in a previous pregnancy, or during the interconception period also carries risk of having a SGA birth in a subsequent pregnancy.

Changes to DNA methylation patterns have been seen in the placentas of women who quit smoking prior to pregnancy and a recent study suggests that tobacco exposure may cause long-term effects via the transmission of epigenetic marks to non-directly exposed placentas [10]. A narrative review of epigenetic alterations due to maternal tobacco smoking in pregnancy concluded that there is increasing evidence to indicate that such alterations persist post-natally, but that there is also the suggestion of some reversibility of DNA methylation when stopping smoking either before or during pregnancy [11].

An analysis of Norwegian Medical Birth Registry data (1999 to 2014) found that daily smoking throughout both of the first two pregnancies was associated with nearly three times the risk of the second child being born SGA (compared to non-smokers in both pregnancies), but that quitting before or during the second pregnancy reduced the risk [12].

We aimed to characterise maternal smoking behaviours across a mother's first two pregnancies and examine the relation of smoking behaviours with the second child's risk of being born SGA. In doing so we examine the hypothesis that mothers who smoked in a previous pregnancy or who smoked between pregnancies have a higher risk of SGA in the second pregnancy compared to never smokers, even if they were not smoking during the second pregnancy. Associations could potentially arise through a variety of biological mechanisms, and these include the effects of smoking on nutritional status or periconceptional development [13, 14]. Whether such

declare. This does not alter the authors' adherence to PLOS ONE policies on sharing data and materials.

**Abbreviations:** ANA, Antenatal appointment; BMI, Body mass index; DAG, Directed Acyclic Graph; GDM, Gestational diabetes mellitus; P1, First pregnancy; P2, Second pregnancy; SGA, Small for gestational age (< 10th percentile); SLOPE, Studying Lifecourse Obesity PrEdictors.

a link is biological or not would depend on how much is it confounded by other factors. This study is observational and so we cannot establish causality, however we believe if such associations were demonstrated this would open the way to exploring possible causal mechanisms.

The exposure groups to be examined include mothers who smoked in their first pregnancies but who quit smoking at some point up to the confirmation of the second pregnancy and those who initiated or resumed smoking after the first antenatal appointment (ANA) for their first pregnancy and reported smoking at the first ANA for their second pregnancy. We also examined non-smokers at the start of both pregnancies but with a history of smoking before one or both pregnancies. Hence, our comparison group was those who never smoked. Identifying women in these groups may enable the targeting of women for interventions.

In addition, we wanted to explore if these relationships are different based on previous history of SGA in the first pregnancy.

## Methods

The SLOPE (Studying Lifecourse Obesity PrEdictors) study is a population-based anonymised cohort of prospectively collected routine antenatal healthcare data collected between January 2003 and April 2018 for women registered with University Hospital Southampton NHS Trust Maternity Services, Hampshire, UK [15–17]. Records for 16791 women with their first two consecutive singleton live-birth pregnancies were included (Fig 1).

This analysis forms part of a research project approved by the University of Southampton Faculty of Medicine Ethics Committee (ID 24433) and the National Health Service Health Research Authority (IRAS 242031).

### Assessment of the exposure

Self-reported smoking status was recorded by a midwife at the first ANA for each pregnancy. For an uncomplicated pregnancy this is recommended to take place by 10 weeks gestation [18]. Women were asked to self-report smoking status at this appointment, and were asked if they were current smokers or if they had ever smoked. If they reported being a current smoker, they were asked how many cigarettes a day they smoked (up to 10 a day/between 10 and 20 a day/more than 20 a day) and the response recorded. Those who reported that they were ex-smokers were asked when they stopped smoking (more than 12 months before conception/ less than 12 months before conception/on confirmation of the current pregnancy).

### Exposure category definitions

A variable was derived to characterise smoking behaviour across the first two pregnancies based on the responses given at the first ANAs for each pregnancy. The full derivation of this variable is given in Table 1.

### Outcome assessment

Age and sex-specific birth weight centiles were used to classify infants born SGA [19]. This was defined as < 10th percentile. Baby's birthweight (grams) was measured and sex was recorded at birth as part of routine care by a healthcare professional. Gestational age (days) was calculated based on a first trimester ultrasound dating scan [18].

### Assessment of covariates

Maternal age (in years) was calculated from date of birth prior to the extraction of the dataset. Maternal weight was measured by a midwife at the first ANA for each pregnancy (kilograms).

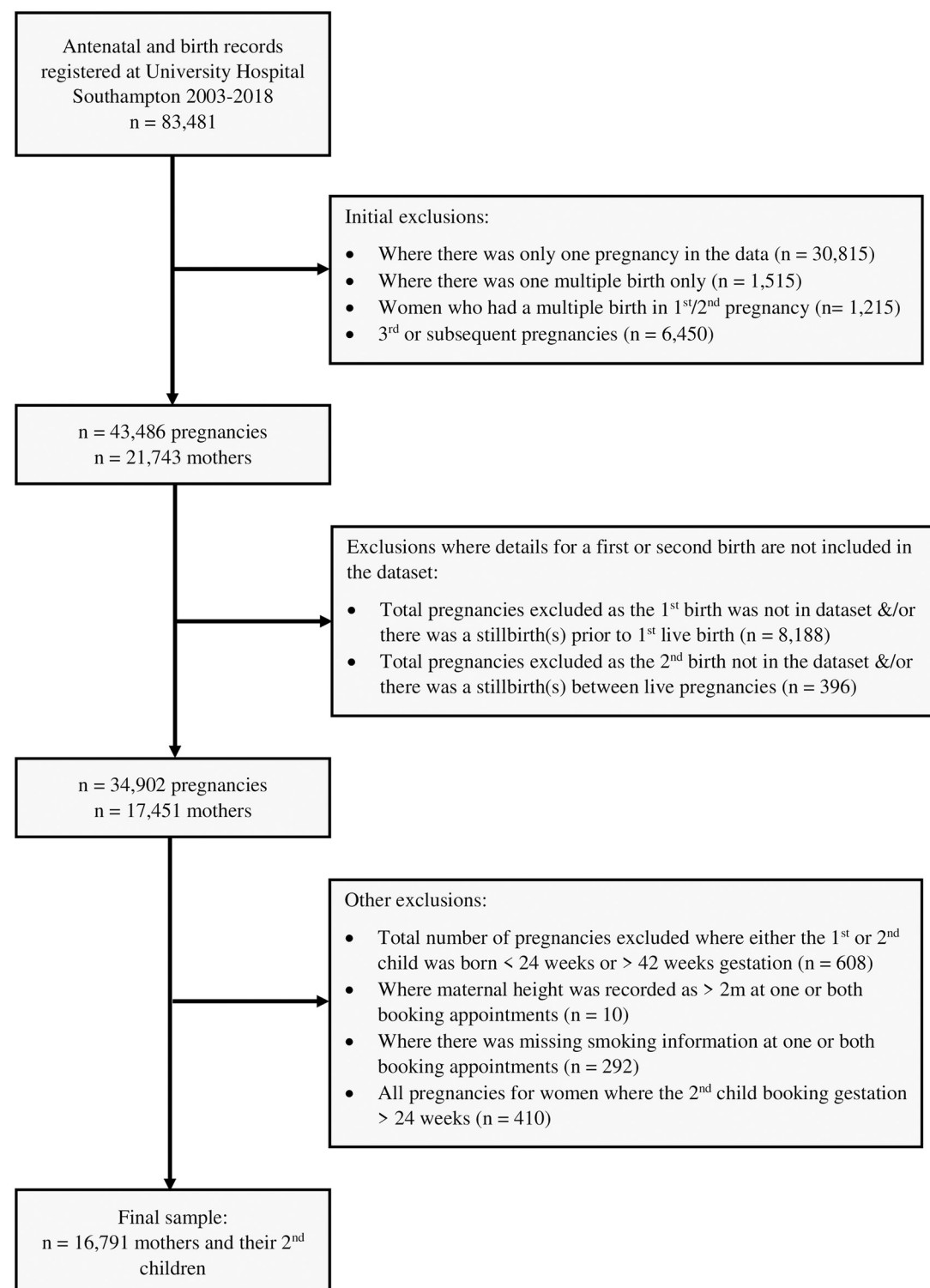

**Fig 1. Flow diagram showing the composition of the final data used in this analysis.** Exclusions from the data are detailed in Fig 1. Births which took place before 24 weeks or after 42 weeks gestation were excluded as SGA reference values do not exist for these gestations. An exclusion for pregnancies where the first ANA for the second pregnancy took place after 168 days gestation (as assessed by ultrasound examination performed by healthcare professionals) was made since these were likely to be high-risk pregnancies referred from elsewhere. Variables documenting the previous numbers of live and stillbirths were used to identify women giving birth for the first and second time and to exclude women who either had a first or second birth elsewhere or who had a stillbirth prior to their first live birth or between live births.

**Table 1. Summary of derived smoking categories based on self-reported maternal smoking status recorded at the first antenatal appointment for each pregnancy.**

| Derived smoking category | Smoking status recorded at first ANA for P1 | Smoking status recorded at first ANA for P2 | Additional notes |
|---|---|---|---|
| Heavier smoker | Smoking 10 or more cigarettes a day | Smoking 10 or more cigarettes a day | These women are the heaviest smokers at the start of each pregnancy |
| Smoker | Smoking up to 10 cigarettes a day | Smoking up to 10 cigarettes a day | |
| Smoker increased | Smoking up to 10 cigarettes a day | Smoking 10 or more cigarettes a day | These women report an increase in the number of cigarettes smoked from the first ANA of P1 to the first ANA of P2 |
| Smoker reduced | Smoking 10 or more cigarettes a day | Smoking up to 10 cigarettes a day | These women report a reduction in the number of cigarettes smoked from the first ANA of P1 to the first ANA of P2 |
| Smoker P2 (not smoking at the first ANA P1) | Not smoking. May be an ex-smoker or have never smoked. If an ex-smoker may have quit at any point up to the confirmation of P1 | Smoking any number of cigarettes | These women may have initiated or resumed smoking at any point after the first ANA for P1 |
| Smoker P1 (stopped before the first ANA P2) | Smoking any number of cigarettes | An ex-smoker who quit at any point up to the confirmation of P2 | These women may have quit smoking at any point after the first ANA for P1; the latest point for cessation would have been on the confirmation of P2 |
| Other smoker (smoker later in P1 or between pregnancies; not smoking at first ANA for P1 or P2) | A non-smoker or an ex-smoker who quit at any point before P1 conception or on confirmation of P1 | An ex-smoker who quit either less than 12 months before P2 conception or on confirmation of P2 | These women did not report smoking at the first ANA for either P1 or P2. They could have smoked later in P1 or after the birth of their first child. They will have smoked at some stage during the 12 months prior to the conception of their second child |
| Ex-smoker | An ex-smoker who quit at any point up to the confirmation of P1 | An ex-smoker who quit more than 12 months before the conception of P2 | These women may have smoked after the first ANA for P1 but did not smoke during the 12 months prior to the conception of their second child |
| Never smoker | Non-smoker with no past history of smoking | Non-smoker with no past history of smoking | |

**Abbreviations**: ANA, antenatal appointment; P1, first pregnancy; P2 second pregnancy.

Height was self-reported (metres) and body mass index (BMI) was then derived ($kg/m^2$). Self-reported variables collected at the first booking appointment for each pregnancy included maternal ethnicity, highest level of educational attainment (secondary (GCSEs) or below/college (A levels)/university degree or above), employment status (condensed to yes/no), partnership status (partnered/lone parent), folic acid supplementation (taking prior to pregnancy/at confirmation of pregnancy/not taking) and infertility treatment (condensed to yes/no). Gestational diabetes mellitus (GDM) and gestational hypertension were identified later during each pregnancy and the diagnosis reported in the database. The interpregnancy interval (days) was calculated based on the World Health Organisation definition [20] by taking the period from the date of the first birth to the conception of the second birth, using the gestational age of the second child. SGA in the first pregnancy was calculated as described in the outcome assessments section above.

## Statistical analysis

Unadjusted comparisons were carried out using Chi-squared tests for categorical variables and ANOVA for continuous variables.

The association between change in smoking behaviour between pregnancies and the risk of SGA birth in the second pregnancy was examined by fitting logistic regression models

predicting a binary outcome (SGA/not SGA). A minimal sufficient adjustment set of confounding variables was identified using a directed acyclic graph (DAG) constructed using DAGitty.net [21, 22] (Fig 2). The DAG illustrates the hypothesised confounding relationships by factors collected at the start of each pregnancy and explicitly identifies our assumptions using *a priori* causal knowledge [21, 23]. References to maternal education and employment in the DAG are taken to be those recorded at the start of the first pregnancy in our analysis.

A large number of minimal sufficient adjustment sets were identified using DAGitty.net [21, 22]. We selected a parsimonious set comprising maternal age, BMI, educational attainment, employment status, partnership status, folate supplementation and infertility treatment details collected at the start of the first pregnancy, diagnoses of gestational diabetes mellitus and gestational hypertension recorded during the first pregnancy, SGA birth in the first pregnancy, maternal ethnicity and the length of the interpregnancy interval (Model 1). The variables were complete in all but 72 cases. In 551 cases ethnicity was not recorded and has been included as "Not specified".

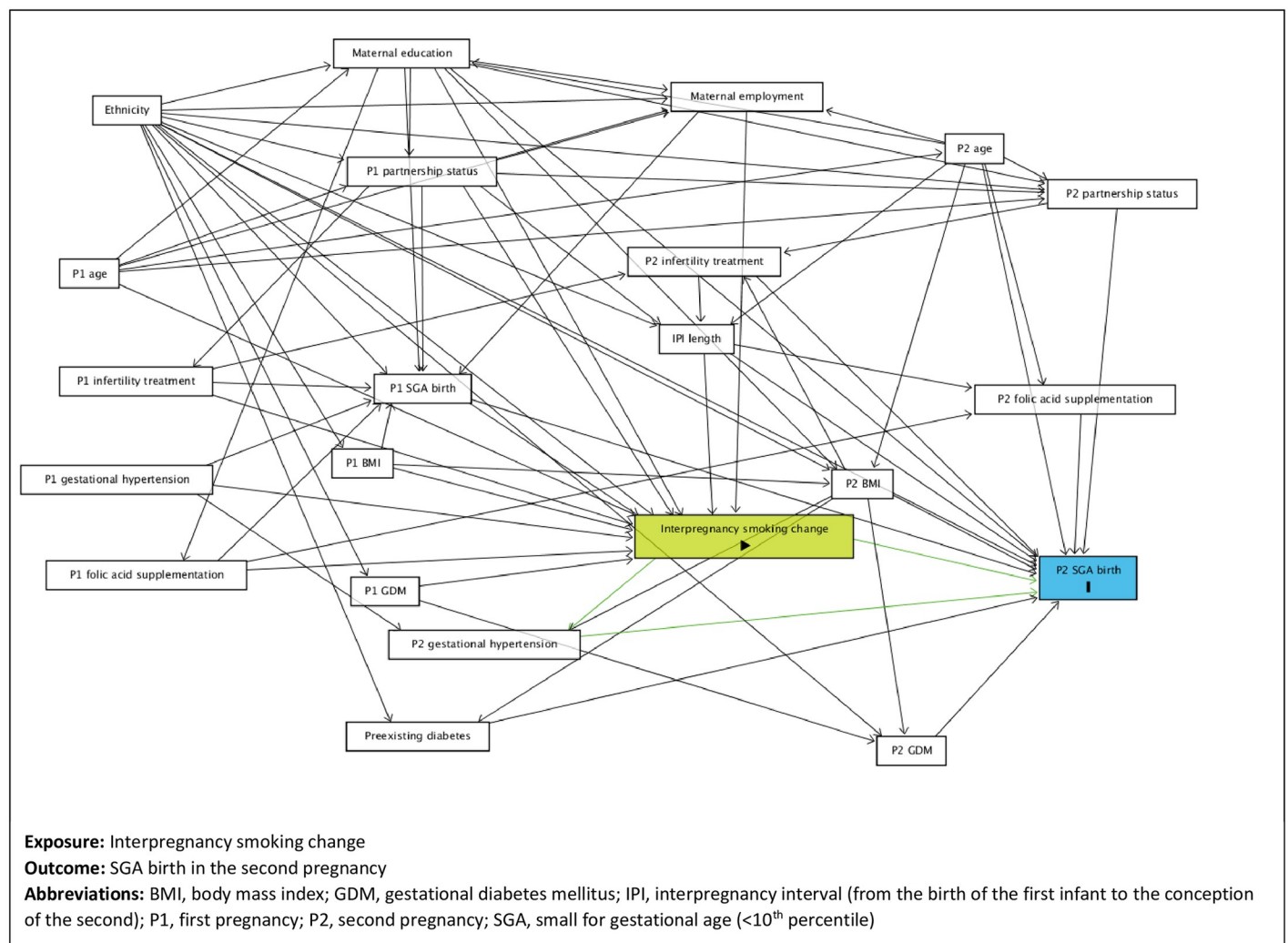

**Exposure:** Interpregnancy smoking change
**Outcome:** SGA birth in the second pregnancy
**Abbreviations:** BMI, body mass index; GDM, gestational diabetes mellitus; IPI, interpregnancy interval (from the birth of the first infant to the conception of the second); P1, first pregnancy; P2, second pregnancy; SGA, small for gestational age (<10th percentile)

**Fig 2. Directed acyclic graph showing the exposure (interpregnancy smoking change) and the outcome (being born small for gestational age (SGA)).**

Each minimal adjustment set identified should close all biasing paths, leaving only measured causal paths open [24]. We used the other sets identified, some of which included covariates collected at the start of or during the second pregnancy, to confirm that there was no change to the results of our analyses and this sensitivity analysis is presented in S1 Table.

Whilst the minimal adjustment set used in this analysis consists mainly of covariates identified at the start of the first pregnancy, a number of second pregnancy covariates may be mediators of the effect of interpregnancy smoking change on SGA birth in the second pregnancy. Analysis was also therefore undertaken to take account of potential mediators. This analysis also followed a minimal adjustment set identified by DAGitty.net [22], but this time taking account of mediators. The identified adjustment set was the same as that identified for Model 1, but with gestational diabetes mellitus and gestational hypertension diagnosed in the second pregnancy in place of that for the first pregnancy and with the addition of maternal BMI recorded at the start of the second pregnancy (Model 2). The adjustment set for Model 2 should close all other measured causal paths with the exception of the effect of interpregnancy smoking change on SGA birth in the second pregnancy [24].

For each Model, analysis was initially undertaken in the whole sample and was then stratified to examine the association with new SGA (where there was no SGA birth in the first pregnancy) and recurrent SGA (following SGA birth in the first pregnancy). Stratified analysis was undertaken on this basis since women who have had a previous SGA birth are known to be at higher risk for a subsequent SGA birth, and therefore previous SGA is hypothesised to be an effect modifier of the effect of smoking on the probability of second SGA [25]. We aimed to assess if the effect estimates are different for the risk of recurrent SGA and new SGA. Our comparison group for all our analyses was never smokers.

All analysis was performed using R [26]. Packages used included data.table [27], dplyr [28], epiDisplay [29], ggplot2 [30], haven [31], psych [32], reshape [33] and tidyr [34].

## Results

Maternal and infant socio-demographics in the second pregnancy, categorised by exposure, are given in Table 2. Of the 16791 women included in this analysis, 49.9% (n = 8386) were categorised as never smokers. There was a slight reduction in the overall percentage of women who reported smoking at the first antenatal appointment for the first pregnancy (15.0%) and the first antenatal appointment for the second pregnancy (13.3%).

Over 70% of women who reported smoking at the first antenatal appointment for their first pregnancy (n = 2522) also reported smoking at the first antenatal appointment for their second pregnancy (n = 1784). Those who smoked at the start of both their first two pregnancies accounted for 10.6% of all included women. A further 4.4% (n = 738) were categorised as smoker P1 (stopped before the first ANA P2) and 2.7% (n = 456) as smoker P2.

Mean maternal age at the start of the second pregnancy was the lowest for all categories of smokers (heavier smokers, (23.8 years, (standard deviation (SD) 4.4)), smokers (24.6 years (4.8)), smoker increased (23.4 years (3.9)) and smoker reduced (24.2 years (4.6)) and smoker P2 (24.7 years (4.8)). Mean maternal age at the second pregnancy was the highest in never smokers (30.2 years (5.0)) and ex-smokers (30.1 years (5.1)). At the start of the second pregnancy and compared to never smokers, all categories of smokers were more likely to be lone parents, of White ethnicity, of lower educational attainment, not to be taking folic acid in early pregnancy, and less likely to be in employment.

In terms of mothers' ethnicity, our sample comprised 86.6% White, 5.9% Asian, 0.6% Chinese, 1.5% Black/African/Caribbean, and 1.2%. Mixed. Other ethnicities comprised 1.0% of the sample and 3.3% did not specify ethnicity.

**Table 2. Maternal characteristics recorded at the first antenatal appointment at the start of the second pregnancy, together with characteristics of both the first and second infants.**

| | Never smoker | Heavier smoker | Smoker | Smoker increased | Smoker reduced | Smoker P2[1] | Smoker P1[2] | Other smoker[3] | Ex-smoker | p-value[4] |
|---|---|---|---|---|---|---|---|---|---|---|
| n | 8386 | 333 | 791 | 347 | 313 | 456 | 738 | 1347 | 4080 | |
| Age, years (mean, SD) | 30.2 (5.0) | 23.8 (4.4) | 24.6 (4.8) | 23.4 (3.9) | 24.2 (4.6) | 24.7 (4.8) | 25.7 (4.7) | 27.1 (5.2) | 30.1 (5.1) | < 0.001 |
| Timing of ANA, weeks (mean, SD) | 11.0 (2.3) | 11.3 (3.3) | 11.2 (2.9) | 11.4 (3.3) | 10.9 (2.6) | 11.0 (2.9) | 11.0 (2.6) | 10.8 (2.5) | 11.0 (2.2) | 0.001 |
| BMI, kg/m$^2$ (mean, SD) | 25.3 (5.3) | 26.1 (6.4) | 25.8 (6.1) | 26.7 (6.5) | 26.5 (6.5) | 26.6 (6.3) | 26.8 (6.1) | 26.7 (6.0) | 26.2 (5.6) | < 0.001 |
| Length of IPI, weeks (median, IQR) | 96 (63, 144) | 91 (46, 164) | 107 (58, 184) | 98 (52, 163) | 113 (58, 189) | 121 (68, 188) | 130 (74, 217) | 123 (74, 190) | 96 (62, 147) | < 0.001 |
| BMI category: | | | | | | | | | | < 0.001 |
| Underweight | 2.9 | 6.6 | 4.9 | 4.9 | 6.1 | 5.0 | 2.2 | 1.8 | 1.8 | |
| Normal weight | 54.9 | 42.9 | 48.4 | 42.1 | 42.5 | 44.5 | 45.0 | 44.0 | 48.6 | |
| Overweight | 25.8 | 26.1 | 24.4 | 25.4 | 26.2 | 23.0 | 27.8 | 29.3 | 29.3 | |
| Obese | 16.4 | 24.3 | 22.3 | 27.7 | 25.2 | 27.4 | 25.1 | 24.9 | 20.3 | |
| Ethnicity: | | | | | | | | | | < 0.001 |
| White | 79.4 | 97.9 | 94.1 | 96.8 | 96.5 | 94.5 | 95.3 | 93.5 | 92.6 | |
| Other ethnicities | 17.1 | 0.6 | 2.8 | 1.7 | 1.3 | 2.6 | 3.0 | 3.6 | 3.6 | |
| Not specified | 3.5 | 1.5 | 3.2 | 1.4 | 2.2 | 2.9 | 1.8 | 2.8 | 3.7 | |
| Highest education level: | | | | | | | | | | < 0.001 |
| University or above | 45.8 | 1.5 | 4.8 | 3.2 | 5.4 | 5.9 | 7.6 | 15.1 | 34.2 | |
| College | 34.9 | 39.6 | 48.3 | 46.1 | 46.3 | 53.3 | 53.4 | 52.3 | 44.9 | |
| Secondary or below | 19.2 | 58.9 | 46.9 | 50.7 | 48.2 | 40.8 | 39.0 | 32.7 | 20.9 | |
| In employment | 72.3 | 28.9 | 41.7 | 33.3 | 38.6 | 44.6 | 55.4 | 64.7 | 73.8 | < 0.001 |
| (missing records) | (n = 57) | (n = 1) | (n = 5) | (n = 2) | (n = 2) | (n = 3) | (n = 2) | (n = 6) | (n = 43) | |
| Taking folic acid: | | | | | | | | | | < 0.001 |
| Prior to pregnancy | 38.5 | 7.2 | 10.7 | 8.9 | 10.2 | 12.1 | 17.2 | 18.9 | 36.2 | |
| At confirmation | 54.3 | 66.7 | 71.4 | 66.6 | 70.6 | 72.6 | 70.9 | 71.1 | 58.3 | |
| Not taking folic acid | 7.1 | 26.1 | 17.8 | 24.5 | 19.2 | 15.4 | 11.9 | 9.9 | 5.5 | |
| Received infertility treatment | | | | | | | | | | < 0.001 |
| Length of the IPI: | 3.7 | 1.2 | 0.4 | 1.7 | 1.3 | 2.4 | 1.6 | 2.3 | 3.3 | < 0.001 |
| < 12 months | 17.2 | 29.4 | 21.1 | 24.5 | 20.4 | 16.7 | 13.7 | 13.2 | 17.8 | |
| 12 to < 24 months | 38.4 | 26.4 | 27.1 | 28.0 | 25.2 | 27.0 | 25.2 | 27.5 | 37.1 | |
| 24 to < 36 months | 23.8 | 15.9 | 20.2 | 19.6 | 21.4 | 20.0 | 19.6 | 23.8 | 23.1 | |
| 36 months or more | 20.6 | 28.2 | 31.6 | 28.0 | 32.9 | 36.4 | 41.5 | 35.4 | 22.0 | |
| Lone parent at P2 | 3.3 | 21.3 | 16.4 | 23.3 | 18.8 | 14.3 | 11.9 | 9.1 | 4.0 | < 0.001 |
| 1$^{st}$ infant birthweight, grams (mean, SD) | 3359.2 (524.0) | 3161.7 (552.2) | 3194.1 (554.4) | 3180.9 (492.6) | 3128.1 (492.8) | 3312.7 (516.1) | 3263.8 (551.3) | 3418.6 (530.3) | 3442.2 (538.2) | < 0.001 |
| 1$^{st}$ infant SGA | 12.0 | 22.5 | 20.6 | 22.5 | 19.8 | 16.4 | 14.9 | 8.6 | 9.4 | < 0.001 |
| 1$^{st}$ infant LGA | 6.6 | 3.6 | 4.0 | 3.2 | 1.6 | 4.4 | 5.7 | 7.9 | 9.2 | < 0.001 |
| 1$^{st}$ infant PTB | 4.9 | 5.7 | 6.4 | 5.8 | 6.4 | 5.5 | 6.4 | 4.2 | 5.0 | 0.253 |
| 2$^{nd}$ infant birthweight, grams (mean, SD) | 3523.8 (511.2) | 3214.4 (544.6) | 3302.6 (535.8) | 3226.1 (534.7) | 3275.5 (505.0) | 3364.9 (530.5) | 3466.9 (551.9) | 3557.8 (538.8) | 3576.2 (512.0) | < 0.001 |
| 2$^{nd}$ infant PTB | 3.1 | 7.8 | 4.6 | 7.2 | 6.4 | 4.6 | 4.1 | 2.7 | 3.3 | < 0.001 |
| 2$^{nd}$ infant SGA | 6.0 | 19.5 | 14.3 | 16.4 | 14.4 | 11.8 | 8.4 | 5.3 | 4.3 | < 0.001 |

(*Continued*)

**Table 2.** (Continued)

| | Never smoker | Heavier smoker | Smoker | Smoker increased | Smoker reduced | Smoker P2[1] | Smoker P1[2] | Other smoker[3] | Ex-smoker | p-value[4] |
|---|---|---|---|---|---|---|---|---|---|---|
| 2nd infant LGA | 13.9 | 6.3 | 6.7 | 4.0 | 5.8 | 7.9 | 13.6 | 15.9 | 15.3 | < 0.001 |

All figures are proportions (%), unless otherwise stated.

[1]. A smoker at the first ANA for P2 who was not smoking at the first ANA for P1.

[2]. A smoker at the first ANA for P1 who stopped before the first ANA for P2.

[3]. A smoker later in P1 or between pregnancies; not smoking at the first ANA for P1 or P2.

[4]. p-values calculated using ANOVA for continuous and Chi-squared tests for categorical variables.

**Abbreviations**: ANA, antenatal appointment; BMI, body mass index; IPI, interpregnancy interval (from P1 birth to P2 conception); IQR, inter-quartile range; LGA, large for gestational age (> 90th percentile); P1, first pregnancy; P2, second pregnancy; PTB, preterm birth (< 259 days); SD, standard deviation; SGA, small for gestational age (< 10th percentile).

The incidence of SGA birth for each of the first two pregnancies by maternal smoking status is shown in Fig 3 and in all cases, the prevalence is lower in the second pregnancy than in the first. The incidence of SGA birth in in never smokers was 12.0% in the first pregnancy and 6.0% in the second pregnancy. For ex-smokers these figures were 9.4% and 4.3% respectively. Of women who have never smoked and who had an SGA birth in the first pregnancy (n = 1004), over a quarter are of Asian ethnicity (n = 257). The incidence of first pregnancy SGA birth for the Asian women included in this study was 27.7%, compared to 11.2% for White women.

Table 4 presents odds ratios for SGA birth in the second pregnancy according to the mother's history of smoking and change in smoking behaviour between the first and second pregnancy, with Model 1 adjusting for confounders, and Model 2 adjusting for confounders and mediators.

Model 1 adjusts for confounders and in the full sample shows the odds of SGA birth in the second pregnancy adjusting for maternal age, BMI, educational attainment, employment status, partnership status, folate supplementation and infertility treatment at the start of the first pregnancy, gestational diabetes mellitus and gestational hypertension recorded during the first pregnancy, SGA birth in the first pregnancy, maternal ethnicity and the length of the interpregnancy interval.

Compared to never smokers, there are increased odds of SGA birth in the second pregnancy for heavier smokers ((adjusted odds ratio (aOR) 3.54 [95% confidence interval (CI) 2.55, 4.92]), smokers (2.44 [1.89, 3.15]), smoker increased (2.70 [1.92, 3.82]), smoker reduced (2.44 [1.68, 3.54]), smokers P2 (2.11 [1.51, 2.95]) and smokers P1 (stopped before the first ANA P2) (1.50 [1.10, 2.03]). Other smokers, (smokers later in P1 or between pregnancies but not smoking at the first ANA of P1 or P2) or ex-smokers did not have increased odds of SGA birth in the second pregnancy compared to never smokers ((1.11 [0.84, 1.45]) and (0.89 [0.73, 1.07]) respectively).

Model 1 in the sample which excludes women whose first child was born SGA makes the same adjustments described above, with the exception of an adjustment for previous SGA birth. Compared to never smokers, there were increased odds of new SGA for heavier smokers (3.53 [2.32, 5.38]), smokers (2.43 [1.75, 3.39]), smoker increased (2.84 [1.82, 4.44]), smoker reduced (2.98 [1.90, 4.67]), smokers P2 (2.22 [1.47, 3.37]) and smokers P1 (stopped before the first ANA P2) (1.75 [1.22, 2.53]). Other smokers (smokers later in P1 or between pregnancies but not smoking at the first ANA of P1 or P2) or ex-smokers did not have increased odds of new SGA compared to never smokers ((1.17 [0.84, 1.62]) and (0.93 [0.74, 1.17]) respectively).

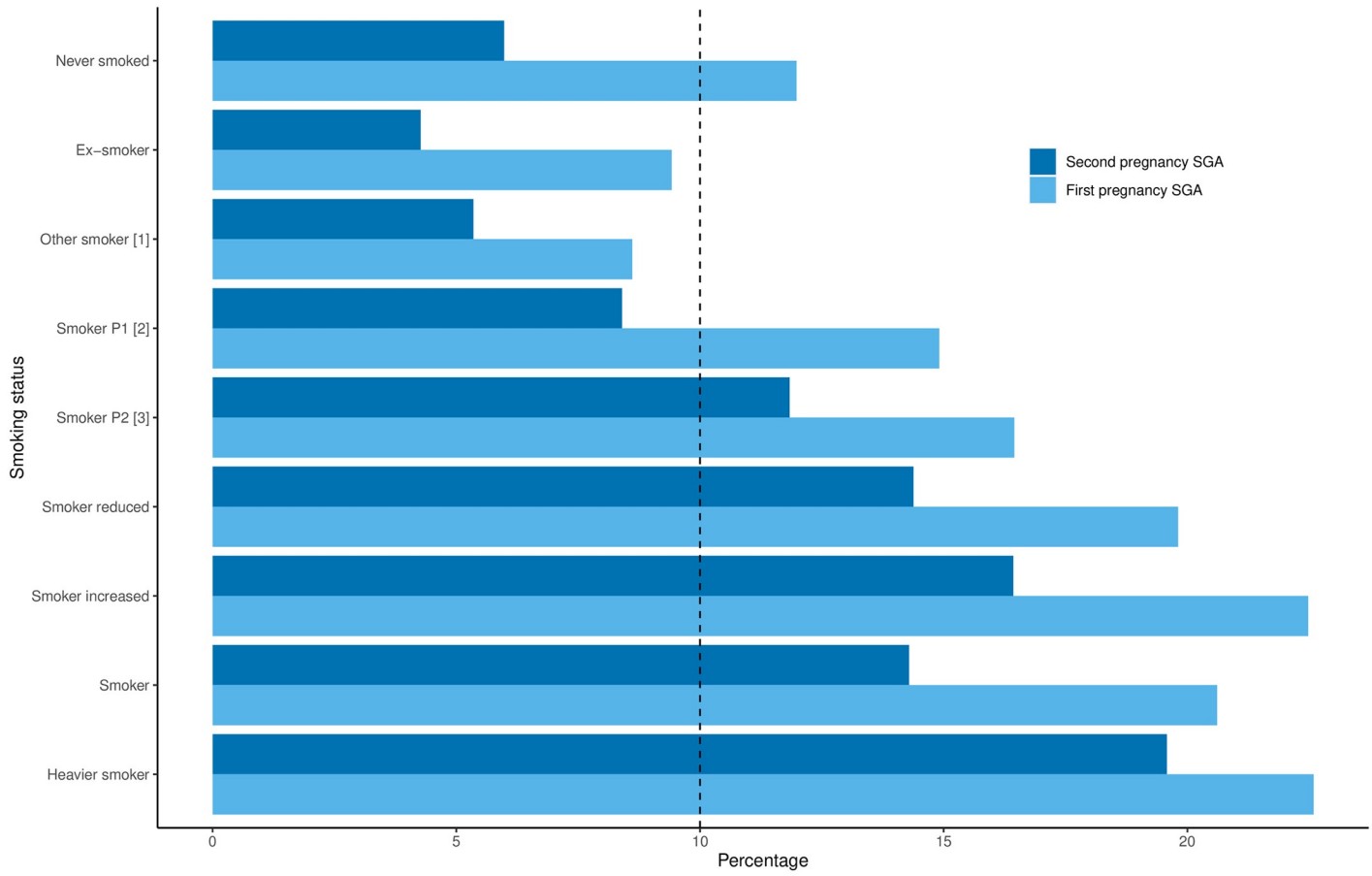

[1] A smoker later in P1 or between pregnancies; not smoking at the first ANA for P1 or P2

[2] A smoker at the first ANA for P1 who stopped before the first ANA for P2

[3] A smoker at the first ANA for P2 who was not smoking at the first ANA for P1

 Abbreviations: ANA, antenatal appointment; P1, first pregnancy; P2, second pregnancy; SGA, small for gestational age (< 10th percentile)

**Fig 3. The percentages of small for gestational age births in the first and second pregnancies.** Table 3 shows the univariate odds of small for gestational age birth in the second pregnancy by maternal characteristics recorded at the start of each pregnancy.

Model 1 in the sample where there was SGA birth in the first pregnancy, shows the odds of recurrent SGA birth. The same adjustments were made as described above.

Compared to never smokers, there were increased odds of recurrent SGA birth in the second pregnancy heavier smokers (3.34 [1.96, 5.68]), smokers (2.34 [1.56, 3.51]), smoker increased (2.42 [1.41, 4.16]) and smokers P2 (1.93 [1.11, 3.36]). Compared to never smokers, there was no increase in the odds of recurrent SGA birth for smoker reduced (1.75 [0.94, 3.25]), smokers P1 (stopped before the first ANA P2) (1.05 [0.62, 1.78]), other smokers (smokers later in P1 or between pregnancies but not smoking at the first ANA of P1 or P2) (0.98 [0.59, 1.64]) or ex-smokers (0.82 [0.59, 1.15]).

Model 2 adjusts for confounders and mediators and in the full sample shows the odds of SGA birth in the second pregnancy adjusting for maternal age, BMI, educational attainment, employment status, partnership status, folate supplementation and infertility treatment at the start of the first pregnancy, gestational diabetes mellitus and gestational hypertension recorded during the second pregnancy, maternal BMI at the start of the second pregnancy, SGA birth in the first pregnancy, maternal ethnicity and the length of the interpregnancy interval.

**Table 3. Univariate odds of small for gestational age birth ($< 10^{th}$ percentile) in the second pregnancy in the full sample, by maternal characteristics recorded at the start of or during the first and second pregnancies.**

| Maternal Characteristics | First pregnancy | | Second pregnancy | |
|---|---|---|---|---|
| | n | OR (95% CI) | N | OR (95% CI) |
| **Age category (ref = 25–34 years)** | | | | |
| < 18 years | 1005 | **2.2 (1.77, 2.73)** | 63 | 1.97 (0.89, 4.33) |
| 18–24 years | 5793 | **1.66 (1.45, 1.89)** | 3913 | **1.65 (1.44, 1.89)** |
| 35–39 years | 845 | 0.99 (0.73, 1.36) | 2386 | 1.01 (0.84, 1.22) |
| 40 years and over | 47 | 1.65 (0.59, 4.60) | 326 | 0.81 (0.49, 1.35) |
| **BMI category (ref = normal weight (18.5–24.9 kg/m$^2$))** | | | | |
| Underweight (<18.5 kg/m$^2$) | 628 | **1.93 (1.51, 2.46)** | 473 | **2.30 (1.77, 2.98)** |
| Overweight (25–29.9 kg/m$^2$) | 4106 | **0.78 (0.67, 0.91)** | 4516 | **0.76 (0.65, 0.88)** |
| Obese ($\geq$30 kg/m$^2$) | 2241 | **0.71 (0.58, 0.87)** | 3282 | **0.69 (0.58, 0.82)** |
| **Highest level of education (ref = degree level)** | | | | |
| College level | 6272 | **1.20 (1.02, 1.41)** | 6923 | **1.31 (1.12, 1.52)** |
| Secondary or below | 5531 | **1.84 (1.57, 2.15)** | 4272 | **1.95 (1.66, 2.28)** |
| **Folic acid status (ref = taking prior to pregnancy)** | | | | |
| Started once pregnancy confirmed | 9624 | **1.55 (1.35, 1.79)** | 9986 | **1.47 (1.27, 1.70)** |
| Not taking folic acid | 1454 | **2.17 (1.76, 2.68)** | 1489 | **2.43 (1.99, 2.98)** |
| Not in employment | 3458 | **2.11 (1.85, 2.40)** | 5530 | **1.91 (1.69, 2.15)** |
| **Received infertility treatment** | 680 | 0.87 (0.63, 1.21) | 514 | 0.75 (0.51, 1.11) |
| **Lone parent** | 1450 | **1.42 (1.17, 1.72)** | 1057 | **1.44 (1.16, 1.78)** |
| **Gestational diabetes mellitus** | 292 | **0.43 (0.22, 0.84)** | 425 | 0.82 (0.54, 1.24) |
| **Gestational hypertension** | 425 | 0.78 (0.51, 1.19) | 188 | **1.74 (1.10, 2.74)** |
| | Non pregnancy specific | | | |
| | n | OR (95% CI) | | |
| **Maternal Ethnicity (ref = White)** | | | | |
| Mixed | 196 | 0.98 (0.55, 1.77) | | |
| Asian | 987 | **2.61 (2.16, 3.15)** | | |
| Black/African/Caribbean | 247 | 1.47 (0.94, 2.29) | | |
| Chinese | 99 | 1.14 (0.53, 2.48) | | |
| Other | 173 | **1.75 (1.07, 2.86)** | | |
| Not known | 551 | 0.93 (0.65, 1.33) | | |
| **Length of the IPI (ref = 12 to < 24 months)** | | | | |
| < 12 months | 2937 | **1.32 (1.10, 1.57)** | | |
| 24 to < 36 months | 3842 | **1.19 (1.01, 1.41)** | | |
| 36 months or more | 4117 | **1.39 (1.19, 1.63)** | | |
| **Previous SGA birth** | 2067 | **6.67 (5.86, 7.58)** | | |

**Abbreviations**: BMI, body mass index; IPI, interpregnancy interval (from the birth of the first infant to the conception of the second); SGA, small for gestational age ($< 10^{th}$ percentile); OR, odds ratio; CI, confidence interval.

Compared to never smokers, there were increased odds of SGA birth in the second pregnancy for heavier smokers (3.57 [2.57, 4.97]), smokers (2.43 [1.88, 3.14]), smoker increased (2.75 [1.94, 3.88]), smoker reduced (2.50 [1.72, 3.63]), smokers P2 (2.13 [1.52, 2.98]) and smokers P1 (stopped before the first ANA P2) (1.53 [1.13, 2.07]). Other smokers, (smokers later in P1 or between pregnancies but not smoking at the first ANA of P1 or P2) or ex-smokers did not have increased odds of SGA birth in the second pregnancy compared to never smokers ((1.12 [0.85, 1.47]) and (0.90 [0.74, 1.08]) respectively).

Model 2 in the sample which excluding women whose first child was born SGA makes the same adjustments described above, with the exception of an adjustment for previous SGA birth. Compared to never smokers, there were increased odds of new SGA for heavier smokers (3.52 [2.31, 5.37]), smokers (2.47 [1.77, 3.44]), smoker increased (2.87 [1.84, 4.49]), smoker reduced (3.05 [1.95, 4.78]), smokers P2 (2.26 [1.49, 3.42]) and smokers P1 (stopped before the first ANA P2) (1.80 [1.25, 2.60]). Other smokers (smokers later in P1 or between pregnancies but not smoking at the first ANA of P1 or P2) or ex-smokers did not have increased odds of new SGA compared to never smokers ((1.17 [0.85, 1.62]) and (0.93 [0.74, 1.17]) respectively).

Model 2 in the sample with SGA birth in the first pregnancy shows the odds of recurrent SGA birth. The same adjustments were made as described above.

Compared to never smokers, there were increased odds of recurrent SGA birth in the second pregnancy heavier smokers (3.54 [2.07, 6.08]), smokers (2.23 [1.48, 3.35]), smoker increased (2.51 [1.45, 4.32]) and smokers P2 (1.91 [1.09, 3.34]). Compared to never smokers, there was no increase in the odds of recurrent SGA birth for smoker reduced (1.82 [0.97, 3.39]), smokers P1 (stopped before the first ANA P2) (1.05 [0.62, 1.79]), other smokers (smokers later in P1 or between pregnancies but not smoking at the first ANA of P1 or P2) (1.01 [0.61, 1.69]) or ex-smokers (0.83 [0.60, 1.17]).

The full results for Model 1 in the full sample (Table 4) are given in Table 5.

Sensitivity analysis for Model 1 was run using the other minimal adjustment sets identified by DAGitty as described in the Methods section above [22]. The results of this analysis (S1 Table) show only very minor differences in the adjusted odds ratios for Model 1 whichever minimal adjustment set is used noting slight differences in the numbers of missing observations across the different models.

## Discussion

In the overall sample we found that mothers smoking at the start of the first pregnancy had a 50% higher risk of SGA birth in the second pregnancy compared to never smokers even if the mother stopped smoking before the first antenatal appointment of the second pregnancy. However, if the mother was not a smoker at the first antenatal appointment for either her first or her second pregnancy, but smoked either later in her first pregnancy or between pregnancies, there was no evidence of increased risk of SGA in her second pregnancy compared to never smokers. When we stratified by previous SGA, this was true for new SGA birth but not for recurrent SGA birth.

According to this analysis, smoking at the start of the first pregnancy may be an important factor in shaping the risk of SGA birth in the second pregnancy. It should be noted, however, that mothers smoking at the start of their first pregnancies could have quit smoking at any point after the first antenatal appointment for their first pregnancy, right up until they found out that they were pregnant for the second time (Table 1).

In all the analyses, second infants born to mothers who reported smoking at the start of both of their first two pregnancies were more likely to be born SGA compared to those of never smokers, with the highest odds of SGA birth found for the heaviest smokers at the start of both pregnancies.

In the analysis of recurrent SGA birth, smokers who reported smoking fewer cigarettes a day at the start of their second pregnancy than they did at the start of their first pregnancy, or who smoked at the start of their first pregnancy but quit by the latest when the second pregnancy was confirmed did not have increased odds of a second infant being born SGA. We do not know however whether these women will have actually quit smoking at some later stage during pregnancy to help avoid a recurrent SGA birth.

**Table 4. The odds of small for gestational age birth (< 10th percentile) in the second pregnancy.**

| | Full sample | | Without previous SGA | | With previous SGA | |
|---|---|---|---|---|---|---|
| | n | Odds Ratios (95% CI) | n | Odds Ratios (95% CI) | n | Odds Ratios (95% CI) |
| **Heavier Smoker** | 330 | **3.79 (2.84, 5.05)** | 256 | **3.66 (2.48, 5.41)** | 74 | **2.66 (1.64. 4.31)** |
| Unadjusted | | | | | | |
| Model 1[†] | | **3.54 (2.55, 4.92)** | | **3.53 (2.32, 5.38)** | | **3.34 (1.96, 5.68)** |
| Model 2[‡] | | **3.57 (2.57, 4.97)** | | **3.52 (2.31, 5.37)** | | **3.54 (2.07, 6.08)** |
| **Smoker** | 786 | **2.64 (2.12, 3.29)** | 623 | **2.48 (1.84, 3.36)** | 163 | **1.93 (1.35, 2.74)** |
| Unadjusted | | | | | | |
| Model 1[†] | | **2.44 (1.89, 3.15)** | | **2.43 (1.75, 3.39)** | | **2.34 (1.56, 3.51)** |
| Model 2[‡] | | **2.43 (1.88, 3.14)** | | **2.47 (1.77, 3.44)** | | **2.23 (1.48, 3.35)** |
| **Smoker increased** | 344 | **3.00 (2.21, 4.05)** | 267 | **2.88 (1.90, 4.37)** | 77 | **1.99 (1.22, 3.25)** |
| Unadjusted | | | | | | |
| Model 1[†] | | **2.70 (1.92, 3.82)** | | **2.84 (1.82, 4.44)** | | **2.42 (1.41, 4.16)** |
| Model 2[‡] | | **2.75 (1.94, 3.88)** | | **2.87 (1.84, 4.49)** | | **2.51 (1.45, 4.32)** |
| **Smoker reduced** | 307 | **2.63 (1.89, 3.67)** | 247 | **3.02 (1.97, 4.61)** | 60 | 1.50 (0.84, 2.65) |
| Unadjusted | | | | | | |
| Model 1[†] | | **2.44 (1.68, 3.54)** | | **2.98 (1.90, 4.67)** | | 1.75 (0.94, 3.25) |
| Model 2[‡] | | **2.50 (1.72, 3.63)** | | **3.05 (1.95, 4.78)** | | 1.82 (0.97, 3.39) |
| **Smoker P2[1]** | 452 | **2.09 (1.55, 2.82)** | 377 | **2.22 (1.50, 3.28)** | 75 | 1.54 (0.92, 2.58) |
| Unadjusted | | | | | | |
| Model 1[†] | | **2.11 (1.51, 2.95)** | | **2.22 (1.47, 3.37)** | | **1.93 (1.11, 3.36)** |
| Model 2[‡] | | **2.13 (1.52, 2.98)** | | **2.26 (1.49, 3.42)** | | **1.91 (1.09, 3.34)** |
| **Smoker P1[2]** | 735 | **1.45 (1.10, 1.91)** | 626 | **1.75 (1.24, 2.46)** | 109 | 0.88 (0.54, 1.44) |
| Unadjusted | | | | | | |
| Model 1[†] | | **1.50 (1.10, 2.03)** | | **1.75 (1.22, 2.53)** | | 1.05 (0.62, 1.78) |
| Model 2[‡] | | **1.53 (1.13, 2.07)** | | **1.80 (1.25, 2.60)** | | 1.05 (0.62, 1.79) |
| **Other smoker[3]** | 1346 | 0.89 (0.69, 1.15) | 1230 | 1.09 (0.80, 1.48) | 116 | 0.82 (0.50, 1.33) |
| Unadjusted | | | | | | |
| Model 1[†] | | 1.11 (0.84, 1.45) | | 1.17 (0.84, 1.62) | | 0.98 (0.59, 1.64) |
| Model 2[‡] | | 1.12 (0.85, 1.47) | | 1.17 (0.85, 1.62) | | 1.01 (0.61, 1.69) |
| **Ex-smoker** | 4065 | **0.70 (0.59, 0.84)** | 3681 | 0.81 (0.65, 1.01) | 384 | **0.65 (0.47, 0.88)** |
| Unadjusted | | | | | | |
| Model 1[†] | | 0.89 (0.73, 1.07) | | 0.93 (0.74, 1.17) | | 0.82 (0.59, 1.15) |
| Model 2[‡] | | 0.90 (0.74, 1.08) | | 0.93 (0.74, 1.17) | | 0.83 (0.60, 1.17) |
| **Never smoker** | 8354 | Reference | 7353 | Reference | 1001 | Reference |

[1]. A smoker at the first ANA for P2 who was not smoking at the first ANA for P1.

[2]. A smoker at the first ANA for P1 who stopped before the first ANA for P2.

[3]. A smoker later in P1 or between pregnancies; not smoking at the first ANA for P1 or P2.

[†] **Model 1 (adjusts for confounders)**: Adjusted for maternal age, BMI, educational attainment, employment status, partnership status, folate supplementation and infertility treatment at the start of the first pregnancy, gestational diabetes mellitus and gestational hypertension recorded during the first pregnancy, SGA birth in the first pregnancy (not in the stratified analysis), maternal ethnicity and the length of the interpregnancy interval.

[‡] **Model 2 (adjusts for confounders and mediators)**: Adjusted for maternal age, BMI, educational attainment, employment status, partnership status, folate supplementation and infertility treatment at the start of the first pregnancy, gestational diabetes mellitus and gestational hypertension recorded during the second pregnancy, SGA birth in the first pregnancy (not in the stratified analysis), maternal BMI at the start of the second pregnancy, maternal ethnicity and the length of the interpregnancy interval.

**Abbreviations**: ANA, antenatal appointment; BMI, body mass index; CI, confidence interval; P1, first pregnancy; P2 second pregnancy; SGA, small for gestational age (<10th percentile).

**Table 5. Full results of Model 1 in Table 4; the adjusted odds of small for gestational age birth ($<10^{th}$ percentile) in the second pregnancy in the full sample.**

| | aOR | 95% CI | |
|---|---|---|---|
| **Maternal smoking status (ref = never smoked)** | | | |
| Heavier smoker | 3.54 | 2.55 | 4.92 |
| Smoker | 2.44 | 1.89 | 3.15 |
| Smoker increased | 2.70 | 1.92 | 3.82 |
| Smoker reduced | 2.44 | 1.68 | 3.54 |
| Smoker P2[1] | 2.11 | 1.51 | 2.95 |
| Smoker P1[2] | 1.50 | 1.10 | 2.03 |
| Other smoker[3] | 1.11 | 0.84 | 1.45 |
| Ex-smoker | 0.89 | 0.73 | 1.07 |
| Maternal age at booking | 1.00 | 0.99 | 1.02 |
| Maternal BMI | 0.98 | 0.96 | 0.99 |
| In employment | 0.83 | 0.70 | 0.97 |
| Lone parent | 0.93 | 0.75 | 1.16 |
| Previous SGA birth | 5.48 | 4.79 | 6.26 |
| Gestational Diabetes | 0.42 | 0.20 | 0.86 |
| Gestational Hypertension | 0.83 | 0.53 | 1.29 |
| Received infertility treatment | 0.88 | 0.62 | 1.26 |
| Length of the IPI (days) | 1.00 | 1.00 | 1.00 |
| **Maternal ethnicity (ref = White)** | | | |
| Mixed | 0.95 | 0.52 | 1.75 |
| Asian | 2.09 | 1.66 | 2.63 |
| Black/African/Caribbean | 1.47 | 0.92 | 2.36 |
| Chinese | 1.37 | 0.62 | 3.05 |
| Other | 1.86 | 1.11 | 3.15 |
| Not known | 1.01 | 0.69 | 1.47 |
| **Maternal education (ref = Degree)** | | | |
| College level | 0.97 | 0.81 | 1.17 |
| Secondary or below | 1.06 | 0.87 | 1.29 |
| **Folic acid (ref = taking prior to pregnancy)** | | | |
| Started once pregnancy confirmed | 1.10 | 0.93 | 1.29 |
| Not taking folic acid | 1.19 | 0.93 | 1.52 |

[1]. A smoker at the first ANA for P2 who was not smoking at the first ANA for P1.

[2]. A smoker at the first ANA for P1 who stopped before the first ANA for P2.

[3]. A smoker later in P1 or between pregnancies; not smoking at the first ANA for P1 or P2.

**Abbreviations**: ANA, antenatal appointment; BMI, body mass index; IPI, interpregnancy interval (from the birth of the first infant to the conception of the second); P1, first pregnancy; P2, second pregnancy; SGA, small for gestational age ($< 10^{th}$ percentile); aOR, adjusted odds ratio; CI, confidence interval.

Maternal smoking is self-reported and there may be an element of under-reporting. Women could either still be smoking at the start of their second pregnancies or resume later during the pregnancy. A comparison of concurrent and retrospective self-reports of smoking status in pregnancy found 19% of all discordant reports (total n = 222) were where mothers recalled smoking daily in pregnancy but had not reported this at the time of their pregnancy and an additional 39% reported occasional smoking where they had registered as non-smokers in pregnancy [35]. The remaining discordant reports were where mothers failed to recall smoking which they had reported in pregnancy [35]. The study found that younger mothers,

multiparae, those with lower levels of educational attainment and those who were not in a stable relationship had lower concordance on reports of smoking in pregnancy compared to older mothers, primiparae, those who were more highly educated and those living with the father at the time of pregnancy respectively [35]. In our study women were asked for their smoking status at the start of each pregnancy and the responses recorded at that time, which means that recall bias is unlikely.

We found similar a similar percentage of women smoked at the start of both of their first two pregnancies to those reported elsewhere [36, 37]. Whilst the time between pregnancies, where a women is still in relatively intense contact with healthcare professionals, is the ideal time to focus on the health of the entire family, particularly for mothers with a previous history of SGA birth, this is obviously a missed opportunity. Whilst mothers who were smoking at the start of their first pregnancy still have an increased risk of SGA birth in their second pregnancy the risk is lower than for those continuing to smoke at the start of the second pregnancy.

Healthcare professionals can refer pregnant smokers to smoking cessation services but there are a number of areas which could be considered and evaluated further. These include smoking support for entire family groups [38]. Financial incentives and rewards have been shown to have a positive impact on increasing long-term rates of smoking cessation in pregnancy and the post-partum period [39]. The use of financial and other interventions, including social media applications, websites and text messaging, have received mixed feedback depending upon whether this was sought from mothers, significant others (including partners) or healthcare professionals [38]. Targeted leaflets, posters and campaigns could be a useful persuasive tool particularly where the specific effects of smoking on the developing fetus are emphasized [38].

## Strengths and limitations

Our study has a number of strengths. The SLOPE study is a large population-based cohort which includes women from all socio-economic and ethnic backgrounds which is representative of the regional population. The ethnic make-up of our sample is comparable with the 2011 England and Wales census with 86% White, 7.5% Asian/Asian British (which includes Chinese), 3.3% Black/African/Caribbean/Black British and 2.2% Mixed/multiple ethnic groups [40].

The Southampton data observatory reports that, based on the 2019 Indices of Deprivation published by the Ministry of Housing Communities and Local Government [41], Southampton is currently ranked 55th out of 317 local authorities based on the average neighbourhood deprivation rank and approximately 45% of the Southampton's population reside in areas which fall within the 30% most deprived nationally [42]. In this analysis approximately half of the women live in Southampton, with half living in the rest of Hampshire which is less deprived.

The analysis was able to adjust for several key confounders and outcome measurements were based on records which were objectively measured by healthcare professionals.

There are some limitations, primarily the fact that the majority of variables were self-reported. Using self-reported maternal smoking status in pregnancy means that there is the possibility of non-disclosure and information bias affecting the ability to characterise the exposure correctly [43]. Suggested methods of overcoming these potential biases are also subject to a number of issues. For example, biologic assays are considered a more accurate way of measuring maternal smoking but still only reflect exposure over short periods and variations in nicotine metabolism affect the net exposure [43].

We were also unable to incorporate risk factors for smoking continuation, inception and cessation such as having a partner who smokes (potentially a different partner to the first pregnancy), other smoking within the household and other exposure to passive smoking.

Repeating this analysis in other datasets will enable the comparison of results to see if our findings are replicated elsewhere.

## Conclusion

In the analysis of the full sample and in women without a previous SGA birth, smoking in the first pregnancy was associated with increased odds of having a SGA infant in the second pregnancy, even if the mother did not report smoking at the first antenatal appointment of the second pregnancy. Where a mother quit smoking at any point up to the confirmation of the second pregnancy, the odds were lower than for women continuing to smoke or those who smoked at the start of their second pregnancy only (compared to never smokers).

In women who were smokers in their first pregnancy and who gave birth to their first infant who was SGA, there was no increase in the odds of having a further SGA infant in the second pregnancy where they quit smoking at any point up to the confirmation of the second pregnancy or where the number of cigarettes a day was reduced from 10 or more in the first pregnancy to up to 10 a day in the second pregnancy (compared to never smokers).

Interventions which support mothers to stop smoking between pregnancies or at the start of her second pregnancy or which help her to reduce the number of cigarettes smoked a day may help to reduce the incidence of having a SGA infant in the second pregnancy.

## Supporting information

**S1 Table. Sensitivity analysis showing the effect of using different minimal adjustment sets on the adjusted odds ratios calculated in Model 1 in the full sample.**
(DOCX)

## Acknowledgments

We thank David Cable (Electronic Patient Records Implementation and Service Manager) and Florina Borca (Senior Information Analyst R&D) at University Hospital Southampton NHS Foundation Trust for support in accessing the data used in this study.

## Author Contributions

**Conceptualization:** Nisreen A. Alwan.

**Data curation:** Nida Ziauddeen.

**Formal analysis:** Elizabeth J. Taylor, Nisreen A. Alwan.

**Funding acquisition:** Nisreen A. Alwan.

**Methodology:** Elizabeth J. Taylor, Nisreen A. Alwan.

**Project administration:** Nisreen A. Alwan.

**Supervision:** Keith M. Godfrey, Ann Berrington, Nisreen A. Alwan.

**Visualization:** Elizabeth J. Taylor.

**Writing – original draft:** Elizabeth J. Taylor, Pia Doh.

**Writing – review & editing:** Elizabeth J. Taylor, Pia Doh, Nida Ziauddeen, Keith M. Godfrey, Ann Berrington, Nisreen A. Alwan.

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
