## [Decision Letter · Decision Letter 0]

15 Jul 2021

PONE-D-21-13251

Maternal smoking behaviour across the first two pregnancies and small for gestational age birth: analysis of the SLOPE (Studying Lifecourse Obesity PrEdictors) population-based cohort in the South of England

PLOS ONE

Dear Dr. Taylor,

Thank you for submitting your manuscript to PLOS ONE. After careful consideration, we feel that it has merit but does not fully meet PLOS ONE’s publication criteria as it currently stands. Therefore, we invite you to submit a revised version of the manuscript that addresses the points raised during the review process.

Enclosed you will find thorough reviews from two experts in the field. They identified major concerns with the manuscript as submitted. Both reviewers and I believe that if the authors are too address the raised points, constituting a major revision, this work would likely meet Plos One's publication criterion. Note major themes of concern were apriori vs post-hoc hypothesis or hypothesis generating approaches, confounding nature of patient self report data, concerns over replicability of the models, co-variate inclusions, and lack of descriptive data.

We look forward to receiving your revised manuscript.

Kind regards,

JJ Cray Jr., Ph.D.

Academic Editor

PLOS ONE

Journal Requirements:

[I have read the journal's policy and the authors of this manuscript have the following competing interests: KMG has received reimbursement for speaking at conferences sponsored by companies selling nutritional products and is part of an academic consortium that has received research funding from BenevolentAI Bio Ltd, Abbott Nutrition, Nestec and Danone. The other authors have no potentially competing interests to declare.].

4. We noted in your submission details that a portion of your manuscript may have been presented or published elsewhere.

[This manuscript is not currently being considered for publication by any other journal but an Abstract based on this analysis was accepted by the Society for Social Medicine and Population Health for their 2020 Annual Scientific Conference and was published in the Journal of Epidemiology and Community Health (Taylor E, Ziauddeen N, Godfrey K, et al OP64 Change in maternal smoking behaviour between two pregnancies and small for gestational age birth: analysis of a UK population-based cohort J Epidemiol Community Health 2020;74:A30-A31.).]

Please clarify whether this conference proceeding was peer-reviewed and formally published. If this work was previously peer-reviewed and published, in the cover letter please provide the reason that this work does not constitute dual publication and should be included in the current manuscript.

6. We note that the grant information you provided in the ‘Funding Information’ and ‘Financial Disclosure’ sections do not match.

Reviewers' comments:

Reviewer's Responses to Questions

**Comments to the Author**

1. Is the manuscript technically sound, and do the data support the conclusions?

Reviewer #1: Partly

Reviewer #2: Yes

2. Has the statistical analysis been performed appropriately and rigorously? 

Reviewer #1: Yes

Reviewer #2: Yes

3. Have the authors made all data underlying the findings in their manuscript fully available?

Reviewer #1: No

Reviewer #2: No

4. Is the manuscript presented in an intelligible fashion and written in standard English?

Reviewer #1: Yes

Reviewer #2: Yes

5. Review Comments to the Author

Reviewer #1: Introduction:

This is an interesting retrospective study from a large prospectively collected dataset. In my opinion its value is as an exploratory study for hypothesis generation. Intuitively, I would have expected that women ceasing smoking between their first and second pregnancies would reduce their risk of SGA in the second pregnancy to around those of ex-smokers or never smokers. Instead, the authors have hypothesised that they have a continued increased risk for SGA when compared to never smokers without clarifying or justifying their hypothesis.

The fact they have shown an increased risk is hypothesis generating but I feel the conclusion is tentative and needs repeating in other data sets. There is significant potential for the result to be biased, based on patient recall of smoking status and the complicated stratification of smoking status that the authors have proposed.

I am also challenged by their decisions regarding covariates on which to adjust the odds ratios. The DAG diagram they present is overly complex and the software used to create a parsimonious set of adjusting covariates is not explained adequately. Their approach here negates any advantage of the DAG process to create a discussion and to explicitly explain and argue for the choices they have made. I would not feel comfortable with the current models without significant additional explanation of their decisions to adjust on covariates present at the start of the first pregnancy not the pregnancy of interest.

The hypothesis:

The stated hypothesis…. “that mothers who smoked in a previous pregnancy or who smoked between pregnancies have a higher risk of SGA in the second pregnancy compared to never smokers, even if they were not smoking during the second pregnancy”. Seems not to be an a priori hypothesis. It seems rather to have come out of the authors analysis of the data. The literature that the authors cite does not support this hypothesis.

I do agree with the authors stated aim… “to characterise maternal smoking behaviours across a mother’s first two pregnancies and examine the relation of smoking behaviours with second child’s risk of being born SGA”.

In my opinion the study would be better characterised as exploratory and hypothesis generating.

The 11 smoking categories for the Exposure are too complicated:

The following are clear and well defined:

• Heavy smoker

• Smoker

• Smoker increased

• Smoker reduced

• Smoker P2

• Ex smoker

• Never smoker

The following are likely to have significant overlap give that the classifications are self reported as well as being subject to recall bias.

• Smoker P1

• Serial quitter

• Smoker later in P1 or between pregnancies

I would like to see the impact of collapsing these into a single category of Smoked between pregnancies; quit prior to or on confirmation P2.

Exploratory analysis of collapsing these groups shows an unadjusted odds ratio for SGA when compared to the never smoked group of OR=1.08; 95% CI [.88-1.31].

As the aim is to examine the relationship between smoking behaviours between pregnancies and the outcomes for P2 whether the following behaviours could also be collapsed to simply things. Smoker – Heavy smoker + Smoker; Not smoking – Ex smoker+ Never smoker. I appreciate that there is a dose-response gradient for the smokers but as the studies stated aim is to examine changes across pregnancies, the inclusion of 11 categories makes the paper complex and more challenging to follow and a more parsimonious approach may help readers.

The use of Directed Acyclic Graph (DAG) technique:

I appreciate that the DAG technique allows for discussion by making underlying relations explicit. However, Figure 2 is incredibly complex and difficult to follow. Further, the use of the software package “DAGitty” to create a parsimonious set of factors to adjust for is poorly explained and has the feeling of a “black box” approach that is opaque to the reader.

At face value I struggle with the author’s or DAGitty’s choice to use variables (Age, Education, Infertility treatment, folic acid) from P1 for the adjusted odds ratio. I struggle even more with the use of complications in the first pregnancy (Gestational diabetes mellitus, Gestational hypertension).

I would appreciate a more explicit explanation of this decision and greater explanation of the DAGitty process and the intervening steps. A more parsimonious diagram of the final proposed causal model would help the reader decide whether they feel the approach and hence any results and conclusions drawn is appropriate.

Table 3:

Could the maternal characteristics at the start of pregnancy 2 be expressed as univariate odds ratios as per Table 6?

This would usually allow the reader to examine whether there is a statistical basis for adjustment on the covariates present at the beginning of the second pregnancy. Notwithstanding that there may be other reasons for inclusion of a covariate based on content knowledge or prior research.

Why did the authors present this table allowing the reader to examine the relationship between maternal characteristics at the start of the second pregnancy when these are not the covariates that they use to calculate the adjusted odds ratios?

Table 5:

In my opinion this sensitivity analysis adds nothing to the study. I would omit it or move it to a supplementary analysis.

Discussion:

As previously discussed, I do not think that there is enough a priori evidence for the authors main hypothesis. Therefore, I feel their conclusion at the beginning of the discussion section is too strong.

This paper is a good exploration of their data set but given the nature of the smoking status variable a sensitivity analysis for all variations of the smoking status that includes any smoking between pregnancies would lead to a null finding.

The authors own discussion acknowledges the issues with patient recall of smoking around pregnancy with discordance rates between 19-39%. Further, it seems that women with higher risk factors for SGA were more likely to have discordance between reported and actual rates of smoking in pregnancy.

Conclusion:

I would recommend a major revision and further explanation of the multivariate models before publication. I would be happy to review a revised draft of this paper.

Reviewer #2: This is a well conducted and described study. I have a few minor comments:

- the counts of individuals in each of the exposure categories - included in the description of the exposure variable in the first column of Table 1 - are results and should not be included in the methods section. They are already included in the first row of table 2 in the results section, so it would be sufficient to just remove them from table 1.

- please remove the ± designation wherever you report SDs and instead report the means followed by the SDs in brackets.

- in Table 2 where you report proportions (e.g. of BMI, ethnicity, education categories), it is unclear what the numbers included in brackets represent. Additionally for the last row where you report n's again, it would be better to instead report the numbers of missing, since the n's are already included in the column headers.

- because the results in Table 2 are purely descriptive (not inferential), you should not be reporting any confidence intervals here. You should simply report mean (SD) or median (IQR) for continuous variables and n (%) for categorical ones (where n is the count of each category and % is the percentage of that count out of the number of individuals in the main exposure/column)

- relating to the comment above, please distinguish clearly between N, the denominator for a proportion, and n the numerator; so for example, you shouldn't have two columns both labelled n as you do in table 3. The first n is the denominator and should be labelled capital N.

6. PLOS authors have the option to publish the peer review history of their article (what does this mean?). If published, this will include your full peer review and any attached files.

Reviewer #1: **Yes: **Adam Mackie

Reviewer #2: No

---

## [Author Response · Author response to Decision Letter 0]

8 Oct 2021

Please see the uploaded response to reviewer comments document and also the cover letter dated 8 October 2021. Please do not hesitate to contact me if I can provide any futher details.

Best wishes

Elizabeth Taylor 

E.J.Taylor@soton.ac.uk

---

## [Decision Letter · Decision Letter 1]

4 Nov 2021

Maternal smoking behaviour across the first two pregnancies and small for gestational age birth: analysis of the SLOPE (Studying Lifecourse Obesity PrEdictors) population-based cohort in the South of England

PONE-D-21-13251R1

Dear Dr. Taylor,

We’re pleased to inform you that your manuscript has been judged scientifically suitable for publication and will be formally accepted for publication once it meets all outstanding technical requirements.

Kind regards,

JJ Cray Jr., Ph.D.

Academic Editor

PLOS ONE

Additional Editor Comments (optional):

Reviewers' comments:

Reviewer's Responses to Questions

**Comments to the Author**

1. If the authors have adequately addressed your comments raised in a previous round of review and you feel that this manuscript is now acceptable for publication, you may indicate that here to bypass the “Comments to the Author” section, enter your conflict of interest statement in the “Confidential to Editor” section, and submit your "Accept" recommendation.

Reviewer #2: All comments have been addressed

2. Is the manuscript technically sound, and do the data support the conclusions?

Reviewer #2: Yes

3. Has the statistical analysis been performed appropriately and rigorously? 

Reviewer #2: Yes

4. Have the authors made all data underlying the findings in their manuscript fully available?

Reviewer #2: No

5. Is the manuscript presented in an intelligible fashion and written in standard English?

Reviewer #2: Yes

6. Review Comments to the Author

Reviewer #2: There are still some results in the methods section. For example, in lines 168-170 you are reporting the number of incomplete observations. You should perhaps instead report what you did with incomplete observations, for example "observations with missing values in these variables were dropped from the analysis. Where ethnicity was not recorded, it was coded as 'not specified'". And then somewhere in the text of the results indicate that 72 observations were dropped for having missing values (if this is what you did). The results tables should also then have the 551 'not coded' observations for ethnicity.

7. PLOS authors have the option to publish the peer review history of their article (what does this mean?). If published, this will include your full peer review and any attached files.

Reviewer #2: No

---

## [Editor Report · Acceptance letter]

9 Nov 2021

PONE-D-21-13251R1 

Maternal smoking behaviour across the first two pregnancies and small for gestational age birth: analysis of the SLOPE (Studying Lifecourse Obesity PrEdictors) population-based cohort in the South of England 

Dear Dr. Taylor:

I'm pleased to inform you that your manuscript has been deemed suitable for publication in PLOS ONE. Congratulations! Your manuscript is now with our production department. 

Kind regards, 

on behalf of

Dr. JJ Cray Jr. 

Academic Editor

PLOS ONE